# PROJECTED COMPRESSION: TRAINABLE PROJECTIONS FOR EFFICIENT TRANSFORMER COMPRESSION

## ABSTRACT

Large language models have steadily increased in size to achieve improved performance; however, this growth has also led to greater inference time and computational demands. Consequently, there is rising interest in model size reduction methods. To address this issue, we propose **Projected Compression**, a novel model compression technique, that reduces model weights by utilizing projection modules. Specifically, we first train additional projection weights and preserve access to all the original model parameters. Subsequently, these projections are combined into a lower-dimensional product matrix, resulting in a reduced-size standard Transformer-based model. Unlike alternative approaches that require additional computational overhead, our method matches the per-token computation cost of training a compressed model. Experimental results show that Projected Compression performs especially well with increasing compression rates as high as 90% compared to other compression methods.

## 1 INTRODUCTION

Large language models (LLMs) are exceptionally good at Natural Language Processing (NLP) tasks (Brown et al., 2020; Raffel et al., 2020; Bommasani et al., 2021). They continue to be developed on an increasing scale, and their computational and memory requirements present growing challenges for deployment, experimentation, and fine-tuning (Cottier et al., 2024; Singh et al., 2025). With reasoning models becoming increasingly popular (OpenAI et al., 2024), their heavy reliance on inference-time scaling poses challenges. To address this, it is essential to reduce the size and operational costs of the model while preserving quality. Doing so will make LLMs more widely accessible to both the research community and the public.

Thus, model compression techniques have emerged as a popular solution for running models more efficiently with constrained resources. Standard hard pruning methods, although very popular and easy to use, suffer from an inherent limitation: once parameters are removed, their representational capacity is permanently lost, often leading to performance degradation.

Given this motivation, we propose Projected Compression (PC), a novel compression method that preserves access to all original model parameters through gradient-optimized projection modules.

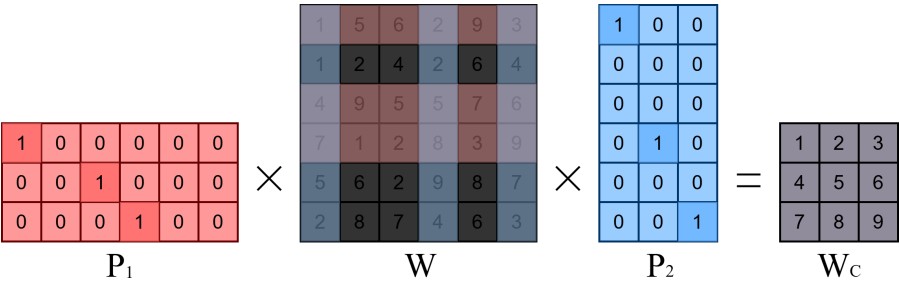

$$P_1 \qquad W \qquad P_2 \qquad W_C$$

Figure 1: Simplified illustration of a projection module, where $P_1$ and $P_2$ are projection matrices, $W$ is a frozen base model parameters and $W_C$ is compressed weights matrix.

Rather than removing unimportant weights, PC redirects their influence through learnable projection modules. Thus, each weight matrix is accompanied with two matrices that produce a lower-rank matrix which is used to build a new compressed model architecture.

Projected Compression gradually builds a compressed representation of the original model, reincorporating and preserving all base model weights throughout the training. Computationally, PC achieves cost per processed token comparable to that of a vanilla Transformer, thanks to projection optimization costs independent of batch size. Therefore, with sufficiently large batches, additional PC costs become negligible.

To the best of our knowledge, our method is the first to reframe model compression as a trainable projection problem operating on a frozen base model that takes advantage of additional weight importance information, explicitly targeting transformer core matrices under structured sparsity constraints.

In summary, the main contributions of this work are:

- Introduction of Projection Compression, a novel compression method that retains access to frozen base weights via trainable projection modules, while maintaining the same training cost per token as standard transformer.
- New possibility of compression of LLMs under high compression ratios, a regime where strong performance is typically difficult to achieve.
- Experimental results which indicate that PC performs better on larger models with stronger pre-training.

Upon acceptance, we plan to release our code to the public as a GitHub repository.

## 2 RELATED WORK

**Pruning.** A widely adopted technique for compressing deep neural networks, model pruning removes weights considered less important for model performance (LeCun et al., 1989; Hassibi & Stork, 1992; Han et al., 2015). Hard pruning methods are based on importance criteria, most commonly magnitude-based thresholds (Han et al., 2015), derivative-based (Hassibi & Stork, 1992) or activation statistics (Hu et al., 2016). Although efficient and simple upon pruning, these methods permanently discard information, limiting their ability to support aggressive sparsity without severe performance degradation.

On the other hand, in soft pruning (Zhu & Gupta, 2017; Mocanu et al., 2020; Evci et al., 2020), the masked weights remain in memory and continue to receive gradients, allowing their reemergence if they regain significance. Dynamic sparse training maintains a fixed number of active connections, reallocating them during training. These approaches outperform static hard pruning, but require complex optimization, specialized sparsity-aware training schedules and may suffer from overhead to irregular parameter usage during compression. More recent methods learn pruning patterns directly through differentiable masking or importance score learning (Xia et al., 2022; Louizos et al., 2017; Wen et al., 2016). Projected Compression similarly enables all weights to influence model compression process mixing their influence by gradient optimized projections.

**Parameter-efficient fine-tuning.** Parameter-efficient fine-tuning (PEFT) approaches, such as LoRA, Prefix Tuning, BitFit, and QLoRA (Hu et al., 2021; Li & Liang, 2021; Ben Zaken et al., 2021; Dettmers et al., 2023), enable model adaptation by freezing the base model weights and introducing a small set of trainable parameters. These methods are designed to efficiently adapt large models to downstream tasks under strict memory or computational constraints, often using lightweight modules such as low-rank adapters. Projected Compression is conceptually related to LoRA and other PEFT approaches by freezing base weights and introducing trainable modules, projection matrices, which adapt the behavior of the projected model without modifying its base model core parameters. Unlike parameter-efficient tuning methods, which adapt models to new tasks, PC is designed as a structured compression strategy: its goal is permanent reduction in model size while maintaining performance, with scaling benefits that improve as base models become larger and better trained. PC does not aim to enable adaptability, but instead to create low-dimensional representations of the full model while maintaining access to its original capacity during training.

**Other work.** In a concurrent work, Hao et al. (Hao et al., 2025) introduced a low-rank clone (LRC) method for compression using low-rank weights that are learned with the use of distillation loss during retraining. Contrary, PC uses weight importance projections initialization similar to dimension reduction operation. Our method is additionally much cheaper matching cost per processed token in retraining with vanilla transformer.

## 3 METHOD

Projected Compression (PC) is a structured compression method that introduces lightweight *projection modules* to build low-dimensional representations of Transformer linear layers as seen in Fig. 1. At the same time, PC retains access to the full set of frozen base parameters as the projection matrices operate over the original weights. Thus, the compressed model is fully defined by these projections, which are gradient optimized during training, while the original base model weights remain frozen.

### 3.1 PROJECTION-BASED STRUCTURED COMPRESSION

Projected Compression targets structured dimension reductions in the Transformer architecture, specifically the *feedforward hidden size* and *embedding dimension* (model width). For each frozen base weight matrix $W \in \mathbb{R}^{d^{\text{in}} \times d^{\text{out}}}$, we are projecting $d^{\text{in}}$ and $d^{\text{out}}$ dimensions to a significantly smaller $d_S^{\text{in}}$ and $d_S^{\text{out}}$. Corresponding projection module is shown in Figure 1. This module consists of one or two trainable projection matrices, $P_1$ and/or $P_2$, which produce a compressed weight $W_C \in \mathbb{R}^{d_S^{\text{in}} \times d_S^{\text{out}}}$ of reduced rank:

$$W_C = P_1 W P_2$$

where:

- $W \in \mathbb{R}^{d^{\text{in}} \times d^{\text{out}}}$ — the frozen base model weights,
- $P_1 \in \mathbb{R}^{d_S^{\text{in}} \times d^{\text{in}}}$ — a projection that downsamples the input dimension,
- $P_2 \in \mathbb{R}^{d^{\text{out}} \times d_S^{\text{out}}}$ — a projection that downsamples the output dimension,
- $W_C \in \mathbb{R}^{d_S^{\text{in}} \times d_S^{\text{out}}}$ — the resulting compressed weights.

During the course of training, only the projection matrices $P_1$ and $P_2$ are updated by gradient descent, while the base weights $W$ remain frozen. This allows frozen weights to be blended into the active computation subspace, offering the potential to recover and leverage information from parameters that would have been permanently removed in standard pruning.

This mechanism preserves compatibility with the standard Transformer architecture. The projected weights $W_C$ are recomputed before each forward pass and are treated as regular layer parameters, ensuring full compatibility with the existing model infrastructure and methods such as an addition of distillation logits loss.

Importantly, the processed token in this construction can be understood as the result of a two-step transformation of the input's dimension: first projected into a higher-dimensional space and using the base model's weights, then reduced back to a lower-dimensional space. Specifically, for a given input token vector $x$, multiplying it by compressed weights $W_C$ is equivalent to first projecting $x$ up into the base model space via $P_1$, performing a matrix multiplication with the full frozen weight matrix $W$, and then projecting the result back down via $P_2$:

$$xW_C = ((xP_1)W)P_2$$

This interpretation highlights that Projected Compression allows each token representation to interact with the full capacity of the base model during training, even though only a compressed projection is used during inference. It reinforces the intuition that PC does not discard any information a priori but instead passes token activations through learnable subspaces of the frozen parameter space.

**One-sided Projection.** While Projection Compression typically employs two projection matrices, it is also possible to use only a single projection matrix. In this one-sided variant, either $P_1$ or $P_2$ is applied, depending on which dimension (input or output) is being compressed.

**Auxiliary Weights Extension.** To improve late-stage flexibility during training, an auxiliary term $W_a \in \mathbb{R}^{d_S^{\text{out}} \times d_S^{\text{in}}}$ is added to the compressed weights:

$$W_C = P_1 W P_2 + W_a$$

The auxiliary weights $W_a$ are initialized with zeros and trained alongside $P_1$ and $P_2$. This addition helps the model maintain the optimization flexibility as it diverges from the frozen base.

## 3.2 TRAINING DYNAMICS AND RESOURCE EFFICIENCY

One of the key advantages of Projected Compression is its training computing efficiency. Forward and backward passes over tokens run through the compressed matrix, $W_C$, not through the full $W$. Gradients w.r.t. activations accumulate in the small parameter space of $W_C$. After this step, those are used to calculate gradients for $P_1, P_2$ via $W$. That operation preserves access to the full model during training but keeps the per-token computations small.

There are two costs for each training step:

1. Fixed costs that do not scale with number of tokens in batch: building $W_C = P_1 W P_2$, calculating $\nabla_{P_1}, \nabla_{P_2}$ from $\nabla_{W_C}$ and applying it.

2. Forward and backward cost through $W_C$ to calculate $\nabla_{W_C}$ which scales with the number of tokens processed in the step (batch).

This separation shows that as we increase the number of tokens per step, the fixed projection cost is amortized and becomes negligible relative to the token-dependent work. Consequently, the total cost per training step is nearly equivalent to that of retraining a compressed transformer model obtained by hard pruning. We formalize the costs of computing derivatives in Projected Compression in the following section.

## 3.3 DIFFERENTIALS AND MATRIX DERIVATIVES FOR PROJECTIONS GRADIENT UPDATE

In the context of Projected Compression, an important question concerns the computational cost of additional weight derivatives. We compute the formula for the derivative of the loss function $L$ with respect to projection matrices, $P_1$ and $P_2$. We show that this derivative can be expressed in terms of the derivative of the loss function with respect to the compressed model weights $W_C$.

Let $L : \mathbb{R}^{d_{\text{in}}^S \times d_{\text{out}}^S} \to \mathbb{R}$. When working with scalar functions of matrix arguments, the differential provides a natural and coordinate-free way to express derivatives. For a scalar loss function $L(W_C)$,

$$dL = \left\langle \frac{\partial L}{\partial W_C}, dW_C \right\rangle,$$

where $\langle U, V \rangle = \text{tr}(U^\top V)$ denotes the Frobenius inner product.

Suppose that we already know the gradient of a scalar function $L$ with respect to $W_C$:

$$G_{W_C} = \frac{\partial L}{\partial W_C}.$$

Since $W_C$ is defined as a multiplication of 3 matrices: $W_C = P_1 W P_2$ it can be also interpreted as the function composition $W_C = g(P_1) = P_1 W P_2$. Then, the differential of $W_C$ is

$$dW_C = dg(P_1) = (dP_1) W P_2.$$

Hence the differential of $L$ can be written as

$$dL = \langle G_{W_C}, dW_C \rangle = \langle G_{W_C}, (dP_1) W P_2 \rangle = \text{tr}\big( G_{W_C}^\top (dP_1) W P_2 \big).$$

Using the cyclic property of the trace, this becomes

$$dL = \text{tr}\Big( \big( G_{W_C} P_2^\top W^\top \big)^\top dP_1 \Big).$$

By definition of the Frobenius inner product, we identify the gradient with respect to $P_1$ as

$$\frac{\partial L}{\partial P_1} = G_{W_C} P_2^\top W^\top$$

In other words, to obtain the gradient with respect to $P_1$, we need to multiply the known gradient with respect to $W_C$ on the right by $P_2^\top W^\top$.

By similar argument, we also obtain,

$$\frac{\partial L}{\partial P_2} = (P_1 W)^\top G_{W_C}.$$

In the one-sided case ($W_C = P_1 W$ or $W_C = W P_2$), the gradients simplify correspondingly to $\frac{\partial L}{\partial P_1} = G_{W_C} W^\top$ or $\frac{\partial L}{\partial P_2} = W^\top G_{W_C}$.

The derivations show that gradients with respect to $P_1$ and $P_2$ can be expressed entirely in terms of the gradient w.r.t. compressed weights $G_{W_C}$, the frozen $W$, and the other projection. This means we do bot need to backpropagate through the full frozen base model $W$. Computationally, the backprop step only involves multiplying by $W$ and $P_2$ (or $P_1$), which is much cheaper than training $W$ itself.

### 3.4 LIMITATIONS

It should be noted that the proposed method comes at the expense of additional memory usage: We store in GPU memory both the trainable projection matrices and the full set of frozen base model weights. This overhead can be mitigated through memory saving techniques such as gradient checkpointing (Chen et al., 2016) and offloading projection matrices to CPU memory during $\nabla_{W_C}$ calculations and load them to GPU memory only for batch size-independent operations. This CPU offloading strategy is particularly effective for projection weights, since they are not involved in computing $\nabla_{W_C}$ and therefore do not require storing the large batch-dependent activations generated by chain-rule forward passes.

## 4 EXPERIMENTS

### 4.1 EXPERIMENTAL SETUP

**Pre-trained models.** As an initial step, we train a series of GPT-2 (Radford et al., 2018) style models to serve as base models for subsequent compression experiments. Specifically, we use two model configurations: a 300M parameter model (16 layers, 16 attention heads) and an 800M parameter model (24 layers, 24 attention heads). In addition to our pre-trained models, we include Llama3 series (Grattafiori et al., 2024) 1B parameters LLM as a baseline model for compression models trained on proprietary data.

**Compression settings.** Multiple compression rates and tokens processed for retraining and Projected Compression were tested using Projected Compression and Hard Pruning with retraining. We have tested random, magnitude-based and activation-based strategies (Sreenivas et al., 2024) to calculate weight importance. Random weight importance was applied to all 800M and 300M models, and to some ablations of the Llama3 1B model compressions. Magnitude importance was used on the ablation experiment 70% with compression of 1B tokens from the Llama3 1B model.

**Evaluation.** To assess the effectiveness of the proposed PC method, we perform a comparative evaluation against the pruning baseline with retraining. The learning rates of each of the training settings were fine-tuned separately for different compression rates and weight-importance methods. Both compression techniques were applied to varying compression setting and performance was measured using the cross-entropy loss averaged over the last 100 training steps. Finally, we evaluate the performance of downstream tasks using the eval-harness framework (Gao et al., 2024).

**Benchmarks.** The AI2 Reasoning Challenge (ARC) dataset provides grade-school science questions with two difficulty levels: ARC-Easy and ARC-Challenge, where the latter requires non-trivial reasoning beyond simple retrieval (Clark et al., 2018). HellaSwag evaluates grounded commonsense reasoning through plausible story continuations (Zellers et al., 2019), while PIQA targets physical knowledge about everyday actions (Bisk et al., 2020). Winogrande is a large-scale pronoun resolution dataset designed to test disambiguation without annotation artifacts (Sakaguchi et al., 2020). Open-BookQA combines a small set of science facts with external knowledge to answer multiple-choice

questions (Mihaylov et al., 2018), and SciQ offers crowdsourced exam-style science questions (Welbl et al., 2017). Social IQa focuses on social reasoning about intentions, reactions, and dynamics (Sap et al., 2019). LAMBADA measures discourse-level understanding by requiring models to predict the final word of passages (Paperno et al., 2016).

**Training setting.** Experiments were conducted across multiple clusters, with compute nodes equipped with 4× NVIDIA GH200, H100, or A100 GPUs. 800M and 300M models were trained and compressed using C4 dataset (Raffel et al., 2020). Llama 3 models were compressed using fineweb-edu dataset (Penedo et al., 2024).

## 4.2 Language Model Loss

The experiments investigate scaling trends by comparing two compression pipelines: Hard Pruning with Retraining (HPR) and Projected Compression (PC). Both methods employ identical weight-importance scores to guide pruning decisions and are computationally matched by being trained on the same number of tokens. As a pruning criterion, we use activation-based pruning (Sreenivas et al., 2024) which we find to be the most competitive among pruning methods.

Table 1 presents the main results that compare our proposed PC method with the hard pruning baseline in three compression rates, 50%, 70% and 90%. Across evaluated configurations, PC consistently yields lower loss than Hard Pruning for higher than 50% compression regimes.

In this work, we also look at two types of models, models trained on open-source and closed-source datasets. While we fully trained the 300M and 800M models, in case of Llama-1B we have no access to proprietary data. In this case, the PC training (or pruning retraining) dataset is different from the pre-training dataset. Hence, we propose dataset alignment procedure where we fine-tune the base model on the target training dataset. Table 1 shows this alignment improves the results for both PC and pruning, however PC benefits more and outperforms pruning even in lower compression rates with sufficient training.

Additional results of smaller models (800M, 300M) compression levels are provided in Table 2, which further confirm the robustness of PC across a range of compression settings with random weights importance method. All of this models conducted experiments can be seen in Section A.2.

| Method | Tokens processed during training | | | |
| | 1B | 2B | 4B | 6B |
|---|---|---|---|---|
| 90% compression | | | | |
| HPR | 3.3870 | 3.2198 | 3.1032 | 3.0500 |
| PC | **3.3155** | **3.1787** | **3.0809** | **3.0366** |
| 70% compression | | | | |
| HPR | 2.9542 | 2.8546 | 2.7767 | 2.7370 |
| PC | **2.9387** | **2.8439** | **2.7700** | **2.7310** |
| 50% compression | | | | |
| HPR | **2.7960** | **2.7005** | **2.6451** | **2.6134** |
| PC | 2.7986 | 2.7038 | 2.6476 | 2.6154 |
| 50% compression with dataset alignment | | | | |
| HPR | **2.7644** | **2.6966** | 2.6495 | 2.6223 |
| PC | 2.7656 | 2.6978 | **2.6493** | **2.6216** |

Table 1: Projected Compression (PC) vs Hard Pruning with retraining (HPR) - cross-entropy loss of Llama3 1B base model. Best loss from each method pair is bolded.

| Model $N$ | Method | Tokens processed during training compression pipeline | | | |
|---|---|---|---|---|---|
| | | 2.5B | 5B | 7.5B | 10B |
| | | 35% compression | | | |
| 300M | HPR | 3.1362 | 3.1070 | 3.0925 | 3.0775 |
| 300M | PC | **3.1361** | **3.1037** | **3.0882** | **3.0711** |
| 800M | HPR | 2.8827 | 2.8519 | 2.8420 | 2.8259 |
| 800M | PC | **2.8780** | **2.8465** | **2.8355** | **2.8192** |
| | | 50% compression | | | |
| 300M | HPR | **3.2172** | 3.1781 | 3.1577 | 3.1402 |
| 300M | PC | 3.2198 | **3.1744** | **3.1528** | **3.1341** |
| 800M | HPR | 2.9733 | 2.9300 | 2.9128 | 2.8930 |
| 800M | PC | **2.9652** | **2.9203** | **2.9037** | **2.8840** |
| | | 65% compression | | | |
| 300M | HPR | **3.3095** | 3.2603 | 3.2392 | 3.2175 |
| 300M | PC | 3.3134 | **3.2591** | **3.2341** | **3.2133** |
| 800M | HPR | **3.0569** | **3.0056** | 2.9828 | 2.9622 |
| 800M | PC | 3.0626 | 3.0069 | **2.9815** | **2.9591** |

Table 2: Projected Compression (PC) vs Hard Pruning Retraining (HPR) - Cross-Entropy loss of base model compression of different sizes (Model $N$) pretrained with 80:1 tokens to parameters ratio. Best loss from each method pair is bolded.

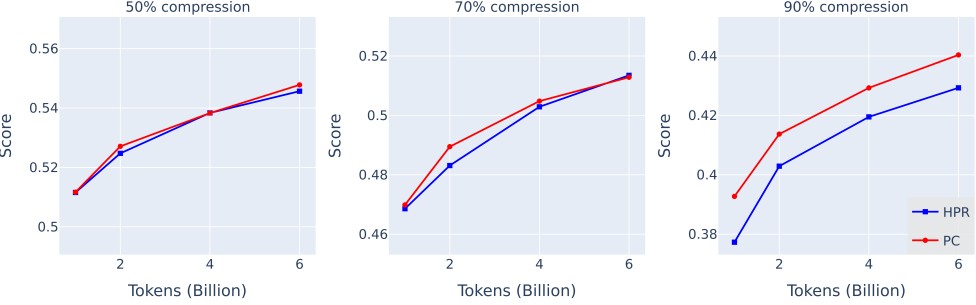

Figure 2: Projected Compression (PC) vs Hard Pruning Retraining (HPR) - Averaged benchmarks for compressed Llama3 1B model. Each data point is obtained from a separate training run. Full down-stream performance for each task can be found in Appendix A.3.

## 4.3 KNOWLEDGE AND REASONING BENCHMARKS

In this section we gauged the performance of Projected Compression on a set of downstream benchmarks. We select a diverse set of tasks to verify if the benefits of PC translate to more diverse tasks.

**Results.** The results for downstream tasks are summarized in Fig. 2. The figures present averaged score over all the downstream tasks to see the overall trend. The detailed results for each of the downstream tasks can be found in Appendix A.3. The experiments show similar trends to the perplexity results. The strong performance of Projected Compression translates to most of the tasks. The edge of Projected Compression is particularly seen with the increase of compression regimes, achieving a substantial difference for 90% compression.

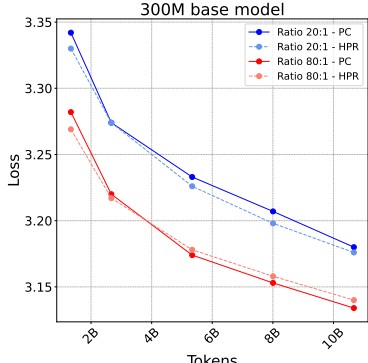 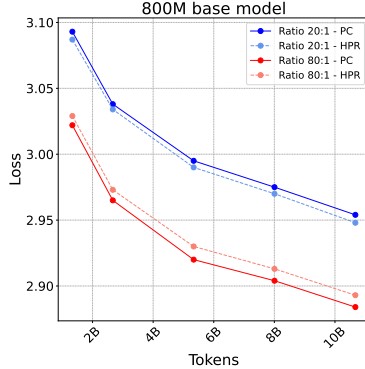

Figure 3: Compression of 50% parameters - Projected Compression (PC) vs Hard Pruning Retraining (HPR) for base models of different sizes pretrained with different tokens to parameters ratio (Ratio $T$:$N$).

## 4.4 Initialization ablation study

In this section, we show that Projection Compression (PC) integrates naturally with other compression methods and can further enhance their performance. In particular, PC can exploit knowledge of important units in the network to guide the initialization of its projection matrices. This ensures that the most important weights dominate the initial projection, while the less important dimensions are initially suppressed.

Nevertheless, unlike pruning, Projected Compression does not eliminate any parameters - less important dimensions are excluded from the initial projection but remain accessible. As training progresses, these suppressed components gradually contribute to the model by optimizing projection weights.

In this study, we provide experimental results about the possible directions of initializing the projection matrices. We ablate over three selection criteria, weight importance based on activations (Sreenivas et al., 2024), weight importance based on magnitude and random initialization. The results are presented in Table 3. First, we note that PC improves on its pruning counterpart. This result shows that PC provides improved capabilities of compression given its design of being able to access the original model architecture throughout the entire training. We note that randomly initialized PC performs better than both magnitude and random pruning setting a strong baseline. At the same time, PC is able to take advantage of stronger initializations and further improve the loss as it is in the case of activation-based importance initialization.

| Initialization Method Comparison | | |
|---|---|---|
| **weight importance calculation method** | **PC** | **HPR** |
| Random | **3.1573** | 3.2546 |
| Magnitude-based | **3.2033** | 3.2453 |
| Activations-based | **2.9387** | 2.9542 |

Table 3: Projected Compression (PC) vs Hard Pruning with Retraining (HPR). Cross-entropy loss for 70% compression of Llama3 1B model with 1B tokens with different weight importance calculation method.

## 4.5 Base model size and quality ablation study

The experimental trends suggest that Projected Compression is particularly well-suited for compressing larger and higher-quality base models. In particular, PC performs best when the model has been trained on a sufficiently large number of tokens relative to its parameter count. This is

illustrated in Figure 3, where Projected Compression yields clear improvements over hard pruning at a token-to-parameter ratio of 80:1. Similarly, the performance advantage of the PC is more pronounced in the 800M-parameter model than in the 300M-parameter variant, as shown in Table 2.

These observations support the view that Projected Compression is a more effective compression method for the types of models that are the most efficient and valuable to compress. This efficiency trend that compression is most beneficial when applied to large, high-quality base models is shown in Frantar et al. work (Frantar et al., 2023).

## 5 CONCLUSION AND FUTURE WORK

In this work, we presented Projected Compression, a novel compression technique that preserves access to all original model parameters through learnable projection weights. Experimental results demonstrate that our method is an effective method for compressing standard transformer-based models. Our results indicate that the effectiveness of Projected Compression relative to hard pruning and retraining increases with base model size, and compression giving the best results at 90% compression ratio. This highlights the potential of PC for strong compression of LLMs, a setting where maintaining performance is typically challenging, as well as its adaptability to a wider range of architectures, which we leave for future exploration.

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

## A  EXPERIMENTS

### A.1  WEIGHT IMPORTANCE IN EXPERIMENTS INITIALIZATION

We have tested the influence on the importance of random and magnitude-based weights for tested base models in both tested compression methods for our 800M and 300M models. The magnitude-based method produced the worst performance than random pruning for each computation method Table 5.

### A.2  ALL EXPERIMENTS

Here, we include results for models trained with lower token-to-parameter ratio where pruning on models trained with a higher token-to-parameter ratio and trained on more tokens. This trend highlights the strength of PC in preserving model quality under data-rich conditions.

| Model size ($N$) | Tokens ratio ($T : N$) | Loss |
|---|---|---|
| 800M | 80:1 | 2.7234 |
| 800M | 20:1 | 2.8547 |
| 300M | 80:1 | 3.0041 |
| 300M | 20:1 | 3.1450 |

Table 4: Base models loss.

| | Projected Compression | | Hard Pruning Retraining | |
|---|---|---|---|---|
| Retraining tokens | 2.5B | 5B | 2.5B | 5B |
| random | **2.9652** | **2.9203** | **2.9733** | **2.9300** |
| magnitude | 2.9655 | 2.9213 | 2.9764 | 2.9320 |

Table 5: Magnitude vs random based weight importance. Cross-Entropy loss when compressing 800M model by 50% (80:1 tokens to parameters ratio).

## A.3 KNOWLEDGE AND REASONING BENCHMARKS

| Model $N$ | Ratio $T{:}N$ | Method | Tokens processed during training compression pipeline | | | | |
|---|---|---|---|---|---|---|---|
| | | | 1.25B | 2.5B | 5B | 7.5B | 10B |
| | | | 35% compression | | | | |
| 300M | 20:1 | HPR | **3.2645** | **3.2175** | **3.1816** | **3.1570** | **3.1339** |
| 300M | 20:1 | PC | 3.2683 | 3.2221 | 3.1845 | 3.1622 | 3.1400 |
| 300M | 80:1 | HPR | **3.1759** | 3.1362 | 3.1070 | 3.0925 | 3.0775 |
| 300M | 80:1 | PC | 3.1776 | **3.1361** | **3.1037** | **3.0882** | **3.0711** |
| 800M | 20:1 | HPR | **3.0074** | **2.9657** | **2.931** | **2.9179** | **2.9031** |
| 800M | 20:1 | PC | 3.0152 | 2.974 | 2.9380 | 2.9238 | 2.9056 |
| 800M | 80:1 | HPR | **2.9224** | 2.8827 | 2.8519 | 2.8420 | 2.8259 |
| 800M | 80:1 | PC | 3.0327 | **2.8780** | **2.8465** | **2.8355** | **2.8192** |
| | | | 50% compression | | | | |
| 300M | 20:1 | HPR | **3.3302** | **3.2741** | **3.2258** | **3.1980** | **3.1758** |
| 300M | 20:1 | PC | 3.3421 | 3.2742 | 3.2329 | 3.2070 | 3.1830 |
| 300M | 80:1 | HPR | **3.2688** | **3.2172** | 3.1781 | 3.1577 | 3.1402 |
| 300M | 80:1 | PC | 3.2823 | 3.2198 | **3.1744** | **3.1528** | **3.1341** |
| 800M | 20:1 | HPR | **3.0872** | **3.0345** | **2.9896** | **2.9695** | **2.9484** |
| 800M | 20:1 | PC | 3.0925 | 3.0380 | 2.9948 | 2.9752 | 2.9543 |
| 800M | 80:1 | HPR | 3.0286 | 2.9733 | 2.9300 | 2.9128 | 2.8930 |
| 800M | 80:1 | PC | **3.0217** | **2.9652** | **2.9203** | **2.9037** | **2.8840** |
| | | | 65% compression | | | | |
| 300M | 20:1 | HPR | **3.4125** | **3.3464** | **3.2919** | **3.2628** | **3.2455** |
| 300M | 20:1 | PC | 3.4241 | 3.356 | 3.3027 | 3.2755 | 3.2523 |
| 300M | 80:1 | HPR | **3.3753** | **3.3095** | 3.2603 | 3.2392 | 3.2175 |
| 300M | 80:1 | PC | 3.3858 | 3.3134 | **3.2591** | **3.2341** | **3.2133** |
| 800M | 20:1 | HPR | **3.1733** | **3.1098** | **3.0577** | **3.0341** | **3.0109** |
| 800M | 20:1 | PC | 3.1848 | 3.1188 | 3.0638 | 3.0380 | 3.0159 |
| 800M | 80:1 | HPR | **3.1245** | **3.0569** | **3.0056** | 2.9828 | 2.9622 |
| 800M | 80:1 | PC | 3.1375 | 3.0626 | 3.0069 | **2.9815** | **2.9591** |

Table 6: Projected Compression (PC) vs Hard Pruning Retraining (HPR) - Cross-Entropy loss of base model compression of different sizes (Model $N$) pretrained with different tokens to parameters ratio (Ratio $T{:}N$). Best loss from each method pair is bolded.

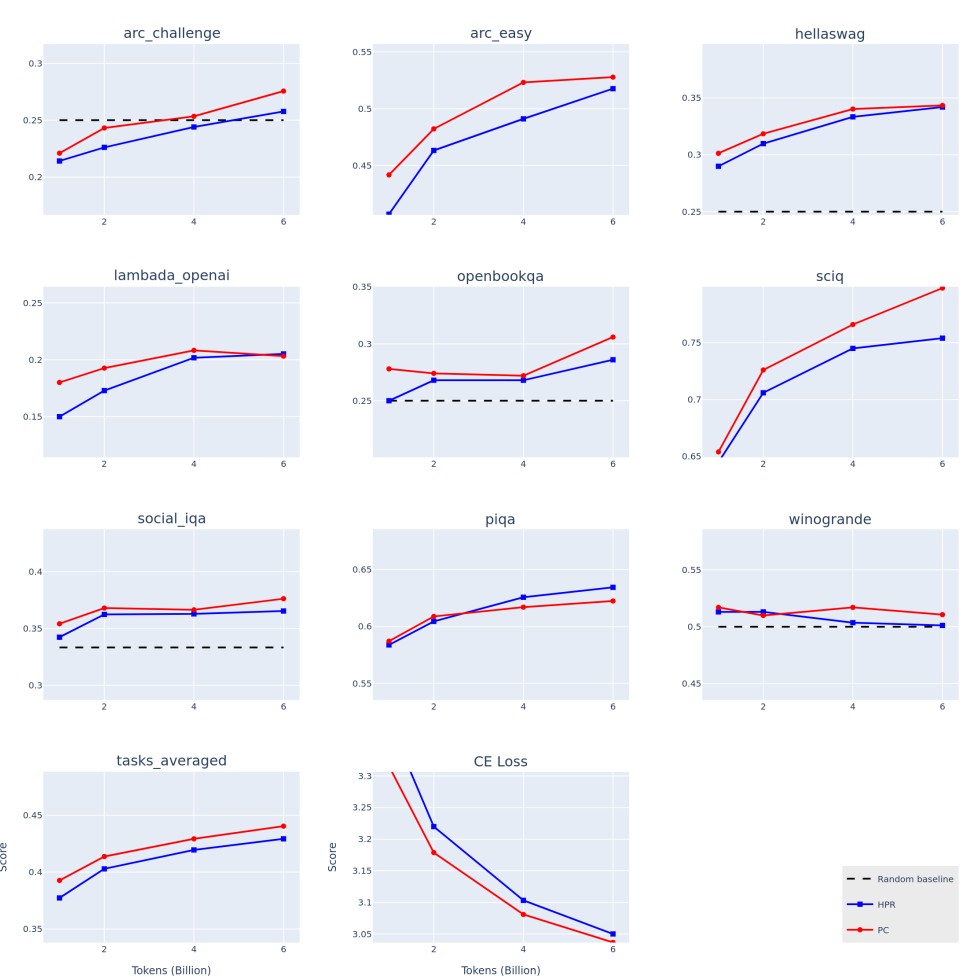

Figure 4: Projected Compression (PC) vs Hard Pruning Retraining (HPR) - Knowledge and Reasoning benchmarks for 90% compressed Llama3 1B model. Each data point is obtained from a separate training run.

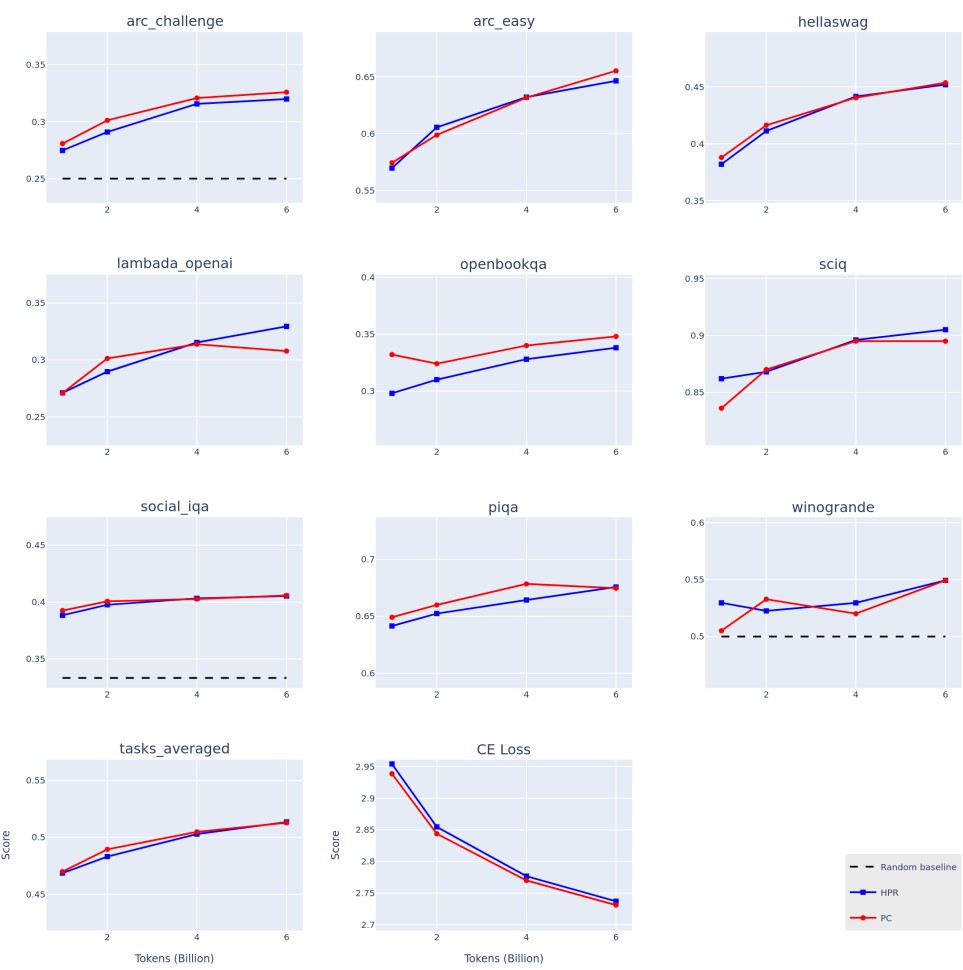

Figure 5: Projected Compression (PC) vs Hard Pruning Retraining (HPR) - Knowledge and Reasoning benchmarks for 70% compressed Llama3 1B model. Each data point is obtained from a separate training run.

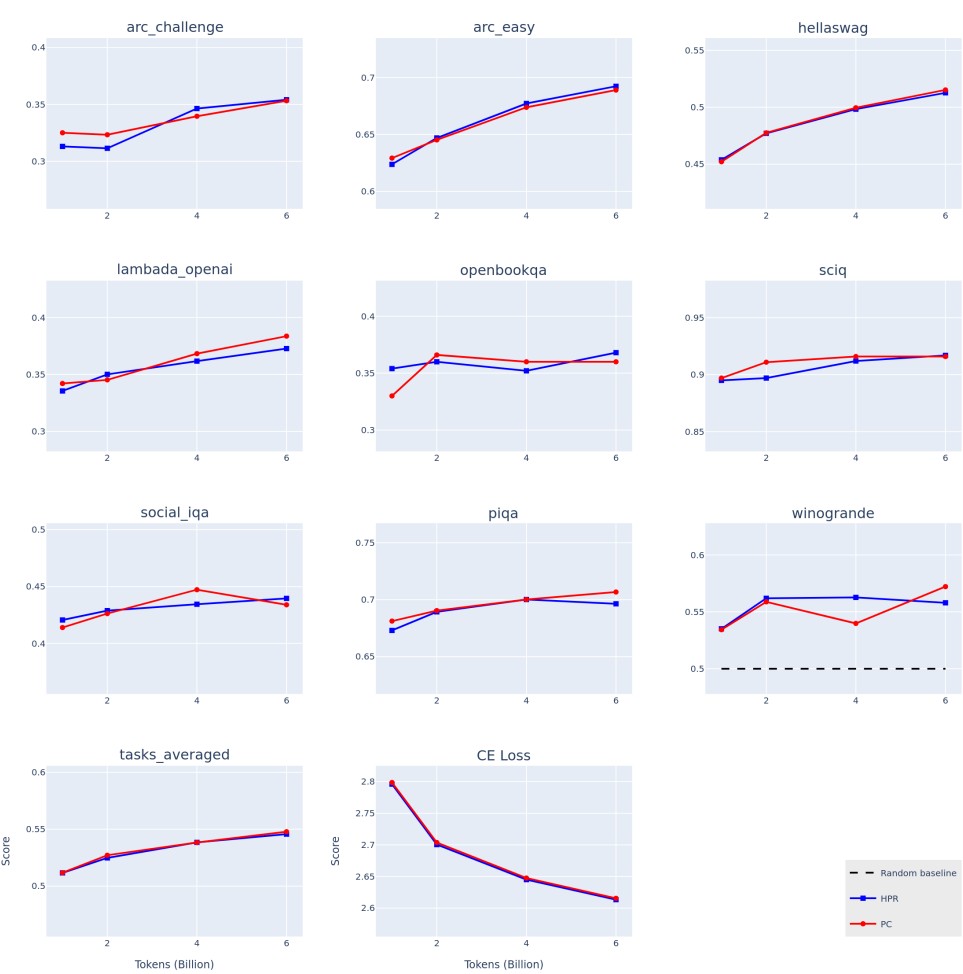

Figure 6: Projected Compression (PC) vs Hard Pruning Retraining (HPR) - Knowledge and Reasoning benchmarks for 50% compressed Llama3 1B model. Each data point is obtained from a separate training run.

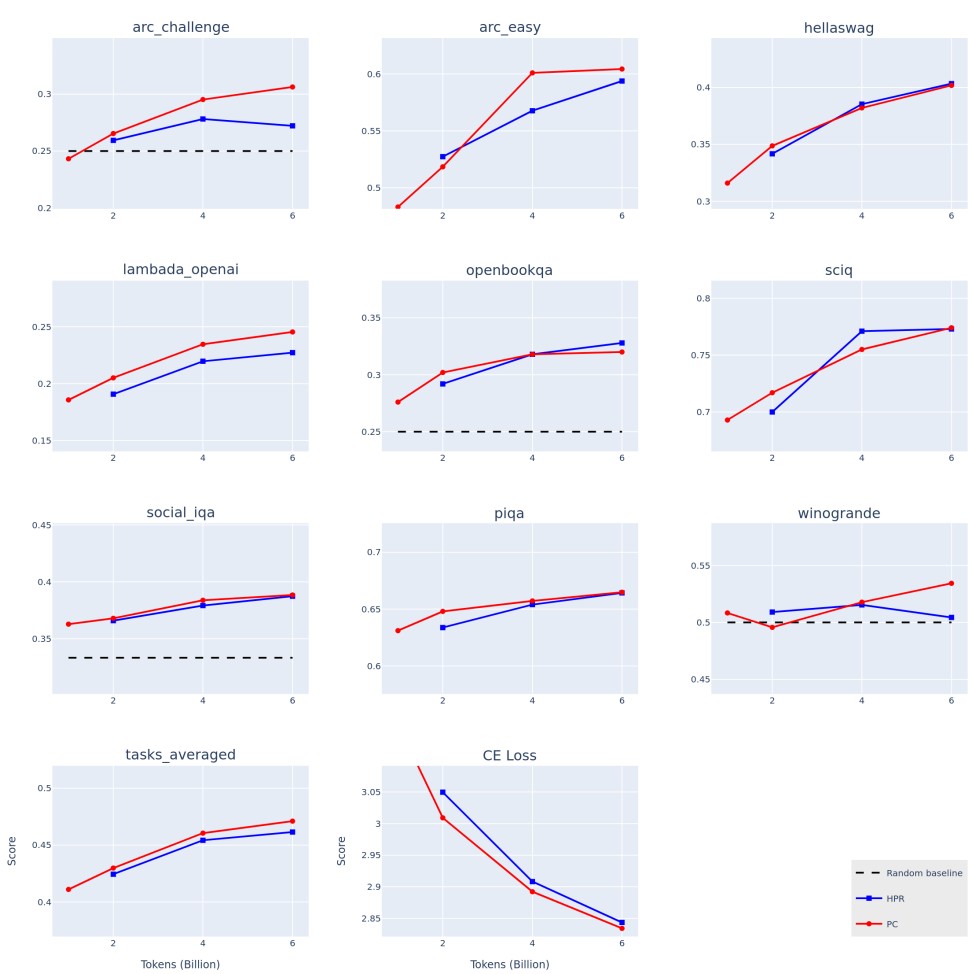

Figure 7: Projected Compression (PC) vs Hard Pruning Retraining (HPR) - Knowledge and Reasoning benchmarks for 70% compressed Llama3 1B model with random weight importance. Each data point is obtained from a separate training run.

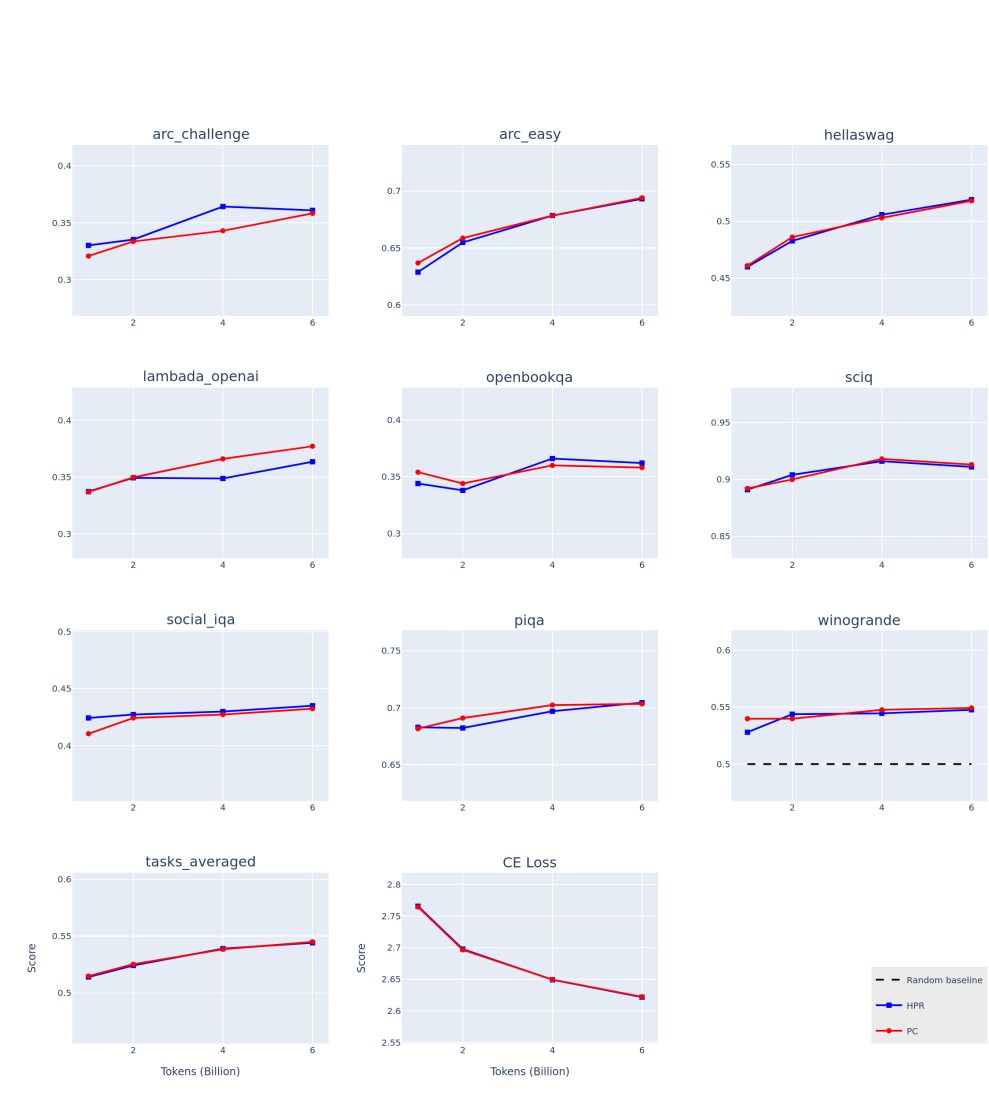

Figure 8: Projected Compression (PC) vs Hard Pruning Retraining (HPR) - Knowledge and Reasoning benchmarks for 50% compressed Llama3 1B model with base model dataset alignment of 10B tokens. Each data point is obtained from a separate training run.

