# OpenReview forum: "Projected Compression: Trainable Projections for Efficient Transformer Compression"
_ICLR.cc/2026/Conference — ICLR 2026 Conference Withdrawn Submission_

### Official Review · Reviewer_ckhy · 2025-10-29

**Soundness:** 3
**Presentation:** 2
**Contribution:** 2
**Rating:** 4
**Confidence:** 3

**Summary:**

This paper proposed Projected Compression (PC), making pruning mask tunable during training and offering a potential to recover and leverage all information of original parameters. Furthermore, authors proved that this approach increased minimal computational cost during continue training, and showed the comparison to hard pruning with retraining.

**Strengths:**

1. method is simple and effective
2. training cost is well estimated

**Weaknesses:**

1. Unclear end-to-end improvement - The baseline (HPR with activation-based importance) is relatively weak. The improvements shown in Tables 1/2 are marginal (typically <0.02 loss difference), which is unconvincing given that PC introduces additional trainable parameters (P1, P2).
2. The paper claims PC can "recover and leverage information from parameters that would have been permanently removed,"(line141).  but provides no analysis or visualization showing:
-  Whether trained P1, P2 actually reactivate initially "pruned" dimensions
- What projection matrices learn compared to initialization
 - Whether the method truly benefits from accessing all original parameters
3. discussions of gradient/PCA based method are missing, where also put more effort on masking.

[1] [ICML2024] LoRAP

[2] [ICLR2024] sheared llama

[3] [ICML2025] SlimLLM

**Questions:**

1. Could you provide evidence that PC retains or recovers more important information after training?


 2. How does PC perform compared to other gradient-based structured pruning methods such as Sheared LLaMA?


3. In Table 3, PC shows strong sensitivity to initialization methods (random: 3.16 vs activation-based: 2.94). If the trainable projections can be optimized during training, why doesn't random initialization converge to comparable performance? Does this suggest limited optimization capacity?

4. Without the auxiliary weights Wa, what are the fundamental advantages of PC's learnable projections over static importance-based masks? Can you provide ablation results comparing PC with and without $W_a$?

---

> ### Author Response · Authors · 2025-12-03
>
> 1. Unclear end-to-end improvement; weak HPR baseline; small loss gains.
> HPR with activation-based importance is the state-of-the-art pruning method. Moreover, we selected pruning as the main benchmark because of the structural similarity which causes a fair comparison to PC: both methods preserve the exact Transformer geometry, both retain the same theoretical FLOPs per token during training, and both achieve the same 1:1 speedup as the compression ratio. Additionally HPR was tested with distillation which we want to incorporate in the next step in our PC research. Most hybrid compression methods outperform its singular-method counterparts, especially with addition of distillation. Under the same FLOPs, the PC outperforms HPR, especially at high compression (e.g., 90%). We will clarify this improvement by adding speedup of achieving similar model loss quality. It is visible when considering distillation addition in our future ablations - when PC benefits from this hybrid configuration more than HPR. It will also be shown in context of speedup given by addition of distillation to HPR and PC.
>
> 2. No analysis demonstrating that PC “recovers pruned information.”; No proof that projections “benefits from access to all original parameters.”
> 	We agree that more direct diagnostics would strengthen the claim. We have tested how PC performs when we occlude by zeros base model weights not selected by projections by initialization initially, and compare it to standard PC to see if access to them speed up convergence. Our results shows that for 90% compression of llama3 1B the loss changes from 3.317 to 3.330 indicating that presence informations pruned by HPR has a positive influence in PC, even when they need to be recovered by trainable projections. Additionally if base model weights (W) are randomly initialize, PC do not converge to a useful state with loss over 7.
>
> 3. Missing discussion of gradient- or PCA-based masking methods (LoRAP, Sheared LLaMA, SlimLLM).
> We will expand the related-work section to contrast PC with these approaches. While such methods rely on static gradient-, PCA-, or saliency-derived masks, PC instead performs anchored optimization in the pretrained weight manifold, where gradients flow through frozen W while P1,P2​ restrict updates to a learnable low-dimensional subspace. This makes PC orthogonal to mask-selection pipelines and complementary rather than redundant. We will clarify this connection.
>
> 4. Sensitivity to initialization (random 3.16 vs activation-based 2.94). Does this imply limited optimization capacity?
> Generally, good neural network initialization prevents exploding/vanishing gradients, speeds up convergence, and makes training stable.In a similar way, (eg. with Xavier and He initialization). In our work, we also performed a similar analysis for the projected compression method and we show that we can further improve the results with some suitable initializations. That being said, even a basic initialization outperforms the baseline. The sensitivity is expected: PC inherits the quality of weight-importance estimation used for initialization (e.g., Wanda-style norms or activation-based methods), similar to HPR. Better initial importance improves early alignment with informative subspaces, accelerating later convergence. Importantly, even random initialization converges to a functional model; it simply starts from a poorer subspace.
>
> 5. Unclear advantage over static importance masks without auxiliary weights W_a​.
> PC without W_a can not converge efficiently at a later stage of training showcasing how projecting manifolds of base model weights can be an anchor in suboptimal optimization space. With W_a​, convergence improves due to increased plasticity.

---

### Official Review · Reviewer_JYSP · 2025-10-29

**Soundness:** 2
**Presentation:** 3
**Contribution:** 3
**Rating:** 4
**Confidence:** 2

**Summary:**

The paper introduces the "Projected Compression" (PC) technique, which compresses a weight matrix by pre- and post-multiplying by trainable projection matrices, resulting in a smaller weight matrix. Unlike PEFT methods like LORA, the PC method has as its goal is permanent reduction in model size while maintaining performance.

**Strengths:**

The training process is efficient, allowing for more rapid learning of the projection matrix weights than for the original large weight matrix entries.

The projection approach better incorporates the original frozen weight matrix information in defining the compressed network than does importance-directed weight pruning of the large network.

The performance of the PC method is significantly better than pruning at high compression rates.

**Weaknesses:**

A minor quibble, but the equations in the paper should be numbered for easy reference.

I would argue with the interpretation that "Projected Compression allows each token representation to interact with the full capacity of the base model during training, even though only a compressed projection is used during inference. " and "It reinforces the intuition that PC does not discard any information a priori but instead passes token activations through learnable subspaces of the frozen parameter space".
The reduction to a subspace means that information IS lost. Even though the subspace is defined by the larger space, all that one has after compression is access to a set of linear combinations of the original weights. Information is lost. All you can hope for is that the information that is lost is not important for the task at hand, which is the same rationale used for weight pruning.
Note that I am not saying that information loss is bad - network compression requires this - just that the interpretation given in the paper that the compressed network still allows "interaction with the full capacity of the base model during training" is suspect. For one, there is an infinity of different large weight matrices (even ones with random entries) that give the same compressed weight matrix, just with different projection matrices. At best, the involvement of the original weight matrix in defining the compressed weight matrix is that of providing a good initialization to the training. To see this, consider just choosing the large weight matrix to be full rank but with random entries. You can still find the projection matrices that will give the same result after training as when using the original weight matrix. So the only potential advantage would be in the speed and quality of the training. To show this, experiments would need to be done to compare training with the original weights to training with random weights. I expect that the original frozen weights would provide a better initialization than the random weights, but this needs to be shown.

Comparison should be made to training the compressed weight matrix from scratch (equivalent to training a smaller model). Normally this does not work as well as start with a large model and then pruning, and so can provide a useful baseline. It also emphasizes the actual benefit of the PC approach as providing a better initialization of the compressed weight matrix.

A drawback of this method as compared with pruning techniques is that a standard back-propagation training phase is required, which can require a large number of iterations of forward and backward passes, resulting in a heavy computational burden. Pruning is typically done via computing weight importance measures based only on single feed-forward passes of the network. No measures of computational expense are provided in the paper.

**Questions:**

What would be the difference if one trained the compression process starting from a randomly initialized large weight matrix instead of the pre-trained large weight matrix?

Were the projection matrices only trained using the benchmark data sets? The original weight matrices were presumably trained with very large pretext datasets, so using only problem-specific datasets to train the projectors means that one is doing fine-tuning only, the the projectors would need to be retrained for every down-stream problem, adding to the effective computational burden.

---

> ### Author Response · Authors · 2025-12-03
>
> 1. Interpretation that PC “interacts with full capacity of the base model” is overstated.
> We clarify that PC does not re-introduce all degrees of freedom of the dense model; rather, it ensures gradients flow through the frozen W before being restricted by the trainable projections:
> ((xP1​)W)P2 ​= x(P1​WP2​)
> Thus, updates can be influenced by all of the base model weights (W) during training, even though the final model retains only the compressed matrix.
> "Projected Compression allows each token representation to interact with the full capacity of the base model during training, even though only a compressed projection is used during inference. "
> This phrasing states that token representation interacts with full base model capacity even though this capacity may be hindered by projecting tokens representation before and after interaction with full frozen base model weights. Because of that it might seem like an overstatement, we will revise it to:
> “ [...] to interact with the all of the base model weights during training [...] “ Thank you for your suggestion.
>
> 2. “PC should be compared to training the compressed weight matrix from scratch.”
> 	We agree and we will add this baseline. From our previous experiment, training an equivalently small model from scratch performs substantially worse as expected.
>
> 3. Additional ablation: What if the base matrix W is random instead of pretrained?
> 	With randomly initialized matrix W, PC converges to a much higher loss. It performs worse than even standard pretraining.
>
> 4. “PC requires backprop; pruning uses only forward passes.”
> 	Typically after pruning we require a retraining phase, which requires backpropagation to update pruned model weights. Our HPR baseline is a hard pruning with a retraining phase afterwards. PC is a method that competes with that approach - FLOPs per processed token during PC training are identical to retraining after hard pruning. Thus, all comparisons were made within the same compute regimes.
>
> 5. “Were projectors trained only on small benchmark datasets? Does this require per-task retraining?”
> 	We have not fine-tuned any model for specific downstream tasks. All methods, including HPR and PC, are evaluated under the same constraints and resources - with the same compute regimes (isoFLOPs) and with the same unsupervised training on FineWeb dataset.

---

### Official Review · Reviewer_nL4b · 2025-10-30

**Soundness:** 1
**Presentation:** 2
**Contribution:** 2
**Rating:** 2
**Confidence:** 4

**Summary:**

This paper introduces Projected Compression (PC), a model compression technique that learns projection matrices to create a compressed weight matrix. Experiments show that PC outperforms traditional hard pruning, especially at high compression rates like 90%.

**Strengths:**

1. LLM weight compression is a critical research direction.
2. Learning projection matrices seems new in this line of research.

**Weaknesses:**

1. The evaluations are insufficient. Only evaluating loss is not a convincing way to evaluate the impact of compression.
2. Also, the compared baselines are limited to Hard Pruning with Retraining (HPR); the paper would be stronger if it compared PC against other modern compression methods.
3. The paper is slightly ambiguous about the final deployed model. It should explicitly confirm that the final artifact is a standard transformer using only the computed $W_C$ weights, with $W$, $P_1$, and $P_2$ discarded.

**Questions:**

See weaknesses.

---

> ### Author Response · Authors · 2025-12-03
>
> 1. “Evaluations are insufficient; only loss is reported.”
> 	We agree that evaluating only loss would be insufficient. However, our submission does report full downstream evaluations on standard reasoning benchmarks (ARC-Easy/Challenge, WinoGrande, HellaSwag, OpenBookQA, PIQA), mirroring the LLaMA evaluation protocol.
> 2. “Baselines are limited; only HPR is used.”
> 	HPR was used as the main baseline because it preserves the exact Transformer geometry via structural pruning, giving a 1:1 speedup–compression ratio and matching PC’s theoretical FLOPs per token. However, we agree that more baselines would strengthen the work. In the revised version of this paper we plan to include results of our method combined with distillation.
> 3. “The final deployed model is ambiguous.”
> 	Thank you for that suggestion. It is worth noting that the text states:
> “Subsequently, these projections are combined into a lower-dimensional product matrix, resulting in a reduced-size standard Transformer-based model.”
> But additional clarification that PC is not a PEFT-style inference-time modification and only a standard compressed weight matrix is used for the inference will be added in the revised version.

---

### Official Review · Reviewer_WgQX · 2025-10-30

**Soundness:** 2
**Presentation:** 3
**Contribution:** 2
**Rating:** 2
**Confidence:** 3

**Summary:**

This paper proposes Projected Compression (PC) that compress models by training two trainable low-rank matrices attached to the frozen weight matrices. This method ensures the access to the original model weights during the training/compression to optimize the compression processes. This paper shows the effectiveness of its methods by conducting experiments on the pretrained models and Llama3-1B model that it consistently outperforms token pruning retraining on different compression ratios.

**Strengths:**

- The writing of this paper is well presented with formulas and graphs showing the workflow of this training algorithm.
- Consistently outperforms the baselines under different settings with limitations discussed about memories

**Weaknesses:**

- *Novelty is limited.* I didn't understand why we need this kind of compression method. A commonly used baseline for the model compressions is directly SVD the original weight matrices to two low rank matrices and train the separate low rank matrices directly. However, I didn't see comparisons over this naive way of compression or any experimental results showing author's method is better than this baseline.
- *The baselines are limited.* There have been many works explores pruning, low rank approximations, as well as other compression methods, such as pruning. In this paper, I only see the prune with retraining with no clear explainations. Authors should compare with more different SoTA methods
- *The improvement over current baselines is limited as well.* From Table 1 and Table 2, the difference between proposed method and the baseline is pretty minimal.
- *Lack of Analysis of convergence speed.*

**Questions:**

My central question of this paper is why we want to design such kind of low rank projection training. If we decompose the original weight matrices into low rank matrices using SVD. The decomposed matrices still have access to the knowledge contained in the original weights. This method should also have more efficient training compared to the current paradigm

---

> ### Author Response · Authors · 2025-12-03
>
> 1. “Novelty is limited – why do we need this method?”
> 	Projected Compression (PC) is not another low-rank factorization method. Unlike SVD-based compression, which fixes a low-rank subspace, PC retains the access to the full pretrained weight matrix during training. The projection matrices learn via gradient optimization how to use the frozen high-rank geometry contrary to SVD which approximates base model weights by linear algebra. This makes the PC a dynamic subspace–learning method, not a static truncation like in the case of SVD.
>
>
> 2. “No comparison with naive SVD low-rank training.”  “Baselines are limited.”
>
> 	SVD was not used mainly as a baseline because unlike PC and HPR, SVD does not give 1:1  speedup–compression ratio, like SVD-LLM yields ~2× weaker speedups and requires more FLOPs per processed token during training. This makes SVD methods weaker competitors to PC if they would require two times shorter training or twice as big sparsity in fixed FLOPs regimes comparison. However, we agree that more baselines strengthen the work. We plan to compare against benchmarks that include knowledge distillation.
>
> 4. “Improvements over current baselines are minimal.”
> 	On higher compression ratios the improvement increases, and on the highest compression ratio improvement is significant and we want to emphasize that it raises with compression ratio.  The differences can be seen even more clearly in the included  experimental evaluation with distillation-based compression.
>
>
> 5. “Lack of analysis of convergence speed.”
> 	Theoretical convergence speed matches our baselines per processed tokens. PC converges faster per token because only the small projection matrices are trainable, but they receive gradients flowing through the entire frozen W. For larger models, this effect strengthens.

---

### Note · Authors · 2025-12-03

**Comment:**

Dear Chairs and Reviewers,

We would like to thank you for the time and effort invested in reviewing our submission. We appreciate all comments, suggestions, and critical insights provided during the evaluation process. The feedback has been valuable in helping us identify the limitations of the current version of the paper, particularly regarding the need for stronger baselines and a more thorough analysis of the proposed method.

After consideration, we have decided to withdraw this version of the manuscript. We are currently working on an improved and expanded version of the work, incorporating additional comparisons, distillation as a part of hybrid compression, and a deeper methodological analysis. We believe this will result in a stronger and more comprehensive contribution, and we hope to resubmit a refined version in the future. Preliminary results show that PC benefits greater from distillation loss than HPR:

### PC vs HPR with distillation loss; llama3 1B compression; CE loss

| Compression | Method | 1B     | 2B     | 4B     | 6B     |
|------------|--------|--------|--------|--------|--------|
| **90%**    | PCD    | **3.295**  | **3.160**  | **3.063**  | **3.020**  |
|            | HPRD   | 3.356  | 3.191  | 3.094  | 3.041  |
| **70%**    | PCD    | **2.896**  | **2.809**  | **2.747**  | **2.716**  |
|            | HPRD   | 2.921  | 2.821  | 2.755  | 2.728  |
| **50%**    | PCD    | **2.739**  | 2.679  | **2.632**  | **2.607**  |
|            | HPRD   | 2.740  | **2.678**  | 2.639  | 2.612  |

Thank you again for your constructive evaluations.

**Withdrawal Confirmation:**

I have read and agree with the venue's withdrawal policy on behalf of myself and my co-authors.